# Increased Pace of Aging in COVID-Related Mortality

**DOI:** 10.3390/life11080730

**Published:** 2021-07-22

**Authors:** Fedor Galkin, Austin Parish, Evelyne Bischof, John Zhang, Polina Mamoshina, Alex Zhavoronkov

**Affiliations:** 1Deep Longevity, Hong Kong, China; poly@deeplongevity.com (P.M.); alex@insilico.com (A.Z.); 2Department of Emergency Medicine, Lincoln Medical and Mental Health Center, Bronx, NY 10451, USA; au.parish@gmail.com; 3Meta-Research Innovation Center at Stanford (METRICS), Stanford University, Stanford, CA 94305, USA; 4International Center for Multimorbidity and Complexity in Medicine (ICMC), Universität Zürich, 8006 Zürich, Switzerland; bischofevelyne@gmail.com; 5Basic and Clinical Medicine Department, Shanghai University of Medicine and Health Sciences, Shanghai 201318, China; 6NYC Health + Hospitals, Lincoln Medical Center, Bronx, NY 10451, USA; John_jz@yahoo.com; 7Insilico Medicine, Hong Kong Science and Technology Park, Hong Kong, China; 8The Buck Institute for Research on Aging, Novato, CA 94945, USA

**Keywords:** aging, biogerontology, COVID, aging clock, prognostics

## Abstract

Identifying prognostic biomarkers and risk stratification for COVID-19 patients is a challenging necessity. One of the core survival factors is patient age. However, chronological age is often severely biased due to dormant conditions and existing comorbidities. In this retrospective cohort study, we analyzed the data from 5315 COVID-19 patients (1689 lethal cases) admitted to 11 public hospitals in New York City from 1 March 2020 to 1 December. We calculated patients’ pace of aging with BloodAge—a deep learning aging clock trained on clinical blood tests. We further constructed survival models to explore the prognostic value of biological age compared to that of chronological age. A COVID-19 score was developed to support a practical patient stratification in a clinical setting. Lethal COVID-19 cases had higher predicted age, compared to non-lethal cases (Δ = 0.8–1.6 years). Increased pace of aging was a significant risk factor of COVID-related mortality (hazard ratio = 1.026 per year, 95% CI = 1.001–1.052). According to our logistic regression model, the pace of aging had a greater impact (adjusted odds ratio = 1.09 ± 0.00, per year) than chronological age (1.04 ± 0.00, per year) on the lethal infection outcome. Our results show that a biological age measure, derived from routine clinical blood tests, adds predictive power to COVID-19 survival models.

## 1. Introduction

The COVID-19 infection was indentified in China at the end of 2019. Since then, it has spread throughout the world, sowing economic turmoil, social unrest, and subjecting national healthcare systems to a harsh test. Despite pandemics having occurred multiple times throughout history, the case of COVID-19 is unique since it is the first pandemic taking place in post-industrial society. A variety of prognostic models were developed to categorize the patients into risk groups, study the factors contributing to poor outcomes, and understand the harmful processes going on during the infection. Previous survival models have shown that older or male patients have a lower hospital discharge probability [1]. Other studies focusing on blood parameters have identified lactate dehydrogenase (LDH) to be the most reliable blood biomarker to predict the infection outcome in severe patients. In a model adjusted for sex, age, treatment, and complications, LDH above 445 U/L has a hazard ratio (HR) of 2 for death [2]. An even lower LDH > 255 U/L has been associated with a 16-fold increase in mortality odds, according to a pooled analysis of nine COVID-19 studies with a total of 1206 infected people [3]. Leukocytosis and hyperglycemia were also identified as significant mortality risk factors [2]. Since the very beginning of the pandemic, COVID-19 has been identified to be a gerolavic (from Greek, géros “old man” and epilavís, “harmful”) infection. In March 2020, people younger than 30 years old accounted for only 0.8% of all COVID-related deaths in China, while the elderly (>60 years) accounted for 81.0% [4]. Aging is a non-stopping damaging process, which reduces resilience towards damaging events, such as COVID-19 infection. However, due to the pace of aging variance, chronological age may not be the best way to quantify this long-term drop in resilience. Biological age is a metric that aims to directly measure the severity of aging-related health issues. There are multiple solutions called aging clocks that can measure biological age based on various biodata types [5]. In the context of the COVID-19 pandemic, aging clocks that can use easily obtained data are the most practical. BloodAge is a neural network aging clock that uses a list of up to 45 blood biomarkers and sex and produces a biological age estimate (see Appendix A) [6]. People who have a higher BloodAge compared to their actual, chronological age are said to exhibit “accelerated aging”. Such people have been shown to have a higher all-cause mortality rates. Additionally, several deleterious behaviors such as smoking are associated with higher BloodAge [7].

In this work, we explored whether biological age is a better predictor of mortality for COVID-19 patients than chronological age. We measured the marginal utility of all available variables to perform feature selection and include only the most important features in our survival model. We hypothesized that an accelerated pace of aging is a significant risk factor even in models corrected for chronological age. To illustrate our findings, we transformed the obtained survival model into a COVID risk score that needs no hardware and can be calculated by hand.

## 2. Materials and Methods

### 2.1. Study Design and Participants

We conducted a retrospective chart review of 11 New York City (NYC) Health and Hospitals (H+H) public hospitals for all adult patients, seen in ED between 1 March and 1 December 2020, who were tested with a polymerase chain reaction (PCR) test for SARS-CoV-2 (COVID-19) during their time in the ED and subsequently admitted. Patients with negative, discontinued, or indeterminate tests were excluded, as were patients that were transferred to hospitals outside of the NYC H+H system. As the same patient may have been presented to the ED multiple times, we used only the earliest visit that resulted in admission so that each patient contributed unique, non-correlated data. We obtained institutional review board (IRB) approval for this study both from Lincoln Medical Center and from the NYC H+H IRB.

We extracted a range of demographic and clinical data for each patient, including initial labs obtained within 24 h of triage. Data were extracted automatically from the Epic electronic medical records (EMR) system. We also excluded patients that had <30 measured values for any of the blood markers required for calculation of biological age (BloodAge).

The total sample of 5315 patients was randomly split into cross-validation (CV, 75%, N = 3987, N_dead_ = 1268) and a verification (25%, N = 1328, N_dead_ = 421) sets.

### 2.2. BloodAge Estimation

BloodAge is an estimate of biological age obtained from clinical blood tests, based on the predictions by the model described in [6].

BloodAge is obtained with a deep neural network that was trained to approximate continuous chronological age based on a vector of up to 46 blood biochemical parameters and donors’ sex. Its output, compared to the patient’s chronological age, represents the intensity of the aging-related changes in a person, compared to same-aged peers. Higher than chronological age, BloodAge values indicate an accelerated pace of aging, while lower ones indicate a decelerated pace (see Appendix A).

The model receives a set of blood variables to produce one value—BloodAge. BloodAge was used to obtain the “Delta age” variable—“underager”, “overager”, and “aging group”:$$\mathrm{Delta}\;\mathrm{age}=\mathrm{BloodAge}-\mathrm{Chronological}\;\mathrm{age}\;Aging\;group=\left\{\begin{array}{c} -1,\;if\;Delta\;age\;<\;-3;\\ 0,\;if\;-3\;\leq \;Delta\;age\;\leq 3;\\ 1,\;if\;Delta\;age\;>3 \end{array}\right.$$
(1)$$Underager\;=\left\{\begin{array}{c} 1,\;if\;Delta\;age\;<\;-3;\\ 0,\;if\;Delta\;age\;\geq -3 \end{array}\right.Overager\;=\left\{\begin{array}{c} 1,\;if\;Delta\;age\;>\;3;\\ 0,\;if\;Delta\;age\;\leq 3 \end{array}\right.$$

### 2.3. Survival Analysis and Feature Selection

Before survival model training, all available blood parameters were transformed into binary variables based on whether the value was below (one) or above (zero) the median in the total COVID sample (see Appendix A). The variables were set to zero in case of missing measurements.

The survival model was an instance of Cox Proportional Hazards (CPH) implemented with lifelines v0.23.9 for Python. CPH models treat available features as independent risk factors and quantify the probability of an event happening by time *t* as:(2)$$h\left(t\right)=h_{0}\left(t\right)\times \exp^{\;HR\times \beta }$$
where $h_{0}\left(t\right)$ is the time-dependent baseline hazard function, HR is a vector of hazard ratios, and β is a vector of independent variables.

To select the most descriptive features, we used a two-step feature selection procedure (see Appendix A).

In the first stage, we used a grid of 9330 different models with a total of 59 variables (<10 independent variables in any model, see Table 1). Each model in the grid contained one of the alternative ways to characterize chronological age (continuous or binary), biological age, obesity, and smoking. Models could include the number of comorbidities and/or symptoms and/or one blood parameter.

Each model was trained with five-fold CV and assigned a concordance index (c-index).

C-index was defined as the number of pair comparisons in which the model guessed the longer survivors based on their expected survival time, relative to the total number of all pairwise comparisons.

All models were ranked according to their average c-index achieved in CV. Each variable was assigned a score—the normalized average rank of all models it was included in. This score belongs in the [0;1] range; higher values indicate a variable’s high significance for accurate survival prediction.

The first stage aimed to remove the most unreliable blood biomarkers and to choose the optimal definition for the variables that allow alternative definitions (e.g., “Never smoker”, “Current smoker”, “Ever smoker” for smoking history correction). Among the 50 highest scoring variables “Never smoker” was the only smoking variable. “Is male”, missing “Is black”, Low P, MCHC, TRIG, BILID, ALP, HGBA1C, BASO%, MCH, HCT, HGB, LDL, CHOLT, PROT, ALT, WBC, BILIT, RBC, GLOBT, HDL, NA+, MCV, CL, and number of comorbidities or symptoms were below the cutoff. These variables were not used in the next round of feature selection.

The passing variables were used to train 26,100 models (Table 1). Each model contained no more than one comorbidity and/or no more than one admission symptom and/or no more than one blood marker. The rank-based scores were used to approximate variables’ marginal utility once again. All variables with a score greater or equal to delta age were included in the final model along with “Never smoker”, “Is male”, and “Is black”.

### 2.4. Adjusted Odds Ratio (AOR)

AOR was defined as the coefficients of the non-regularized LogisticRegression fitter from the sklearn.linear_model v 0.22.1 for Python. Only the variables present in the final survival model were tested. Standard deviations of AORs were calculated based on five-fold CV. Censored entries were considered survivors.

### 2.5. Survival Classifier

The final CPH model was transformed into a binary classifier that would predict a patient’s survival status on a timeframe ranging from one to 130 days (only 23 patients were observed for >130 days) using their median survival function value as the cutoff (see Appendix A).

The most effective classifier timeframe was defined as the convergence point of the sensitivity and specificity curves. Sensitivity was defined as the number of true positive predictions relative to all positive samples, while specificity was defined as the number of true negative predictions relative to all negative predictions (see Appendix A).

### 2.6. COVID Score Composition

A COVID-19 risk score was developed to classify people into four groups based on the expected time to death. The lowest coefficient in the model (−0.53 for below-median LDH) was multiplied by ten and rounded to the nearest integer (−5). All other coefficients were scaled relative to LDH’s weight. The score is the sum of all such coefficients, which is shifted so that the zero score indicates the lowest possible mortality risk. The score’s maximum possible value is 55.

Censored entries were considered survivors when the score was tested as the lethal outcome predictor.

Details of the score composition are available in Appendix A.

### 2.7. Code Availability

The study does not include any novel mathematical models and can be reproduced using publicly available Python packages. The final survival model (CPH fitter object, as implemented in lifelines v.0.23.9) is planned to be released for public use before publication. The COVID risk score is also planned to be released for public use as a website application. BloodAge, the deep learning model used to obtain biological age estimates, is publicly available for academic use at http://www.aging.ai (accessed on 19 June 2021) and consumer or commercial use at http://www.young.ai (accessed on 19 June 2021).

The reported COVID risk score is available online at https://app.young.ai/covid (accessed on 19 June 2021). This application is also available as a standalone: https://cherrypy.org/ (accessed on 19 June 2021) project at Open Science Framework https://dx.doi.org/10.17605/OSF.IO/T6VGD (accessed on 19 June 2021).

## 3. Results

### 3.1. Study Sample

Between 3 January 20 and 12 January 20, a total of 82578 adult patients were tested for COVID-19 in the emergency departments (EDs); of these tests, 12902 (15.6%) were positive. Of these patients, 3377 (26.2%) were discharged home, 487 (3.8%) were transferred to another facility outside the hospital system, 129 (1.0%) left against medical advice, 272 (2.1%) died before admission, and 8637 (66.9%) were admitted. Of these, 8510 (98.5%) represented a unique patient admission. Of these unique patients, 263 (3.1%) died within 48 h of triage and were excluded. Of the remaining 8247 patients, 2932 were excluded due to missing values in any of the non-blood variables or in case they had <30 blood parameters among those required for BloodAge calculation. This left a total of 5315 patients for the primary analysis, among them being 1689 lethal cases.

### 3.2. Accelerated Aging as a Mortality Risk in COVID-19

In the COVID data collection, comprising 5315 patients, BloodAge displayed a mean absolute error (MAE) of 2.80 years (Figure 1A). The patients that died were predicted on average to be 0.99 years older than the group of the censored survivors (Table 2). Males, in general, were predicted to be 0.38 years older than females. COVID patients that died were predicted to have a higher biological age than patients that survived—by 0.97 and 0.93 years in males and females, respectively. Across patients that survived, males were predicted to be on average 0.40 years older than females. For patients that died, biological age was not significantly different between males and females.

Delta age decreased with chronological age from +3.34 years on average in the 20–39-year-old group to −2.64 years in the 80–99-year-old group (Figure 1B, see Appendix A). In the meantime, in each age group, except for those aged 80–99 years, lethal cases were predicted to be significantly older. Lethal cases in the 20–39 age group were predicted to be 1.61 years older, in the 40–59 group—1.47 years older, and in the 60–79 group—0.78 years older.

### 3.3. Biological Aging in Survival Models

We tested BloodAge in the context of survival models corrected for demographic factors, health conditions, and blood parameters. To choose between the alternative ways to define the model, we used a grid search procedure. In it, each variable considered for inclusion was scored according to the average c-index of all the models it was a part of (see Appendix A). The final CPH model contained corrections for seven blood parameters (BUN, creatinine, LDH, and relative eosinophil, lymphocyte, monocyte, neutrophil counts), sex, race, chronological and biological age (delta age), two comorbidities (diabetes, smoking), and two admission symptoms (altered mental state—AMS, dyspnea)—see Table 3.

The point estimate for delta age hazard ratio (HR) was 1.026 (95% CI 1.001–1.052) per year in the presence of chronological age correction (Figure 1C). The model reported reached a c-index of 0.748 in the training set and 0.743 in the test set. No significant difference in accuracy was detected when all binary variables were switched for their continuous versions: c-index = 0.752 in the training set, c-index = 0.742 in the test set.

### 3.4. Survival Classifier

We reworked the CPH model into a survival status classifier. A patient’s median survival time was used as a cutoff to determine if they were likely to survive for at least T days after admission. A range of T from one to 130 days was tested in the verification sample (1328 patients, including 421 deaths). The classifier reached a maximum performance at T = 18 days: 62% specificity and 61% sensitivity (Figure 1D).

### 3.5. Adjusted Odds Ratio (AOR)

We used the same 15 features for the AOR analysis to see if they are predictive of the outcome with the time dimension omitted.

Delta age (AOR = 1.09 ± 0.00, per year) was deemed to have more impact on mortality than chronological age (AOR = 1.04 ± 0.00, per year).

The resulting logistic regression of the COVID-19 infection outcome yielded 57% sensitivity and 89% specificity in the verification set of 1328 patients (Table 4).

### 3.6. COVID Risk Score

We propose a COVID-19 mortality risk score based on the CPH model that can be calculated manually (Table 5). The score is a linear sum of the normalized non-exponentiated CPH coefficients. Its minimal value of zero translates into 236 days expected survival time, the maximum score of 55 translates into four days (see Appendix A).

We propose classifying patients into four risk groups: low risk (0–21 points, expected survival >134 days), moderate risk (22–34 points, >38 days), high risk (35–41 points, >14 days), critical risk (42–55 points, ≤14 days).

Within the verification sample of 1328 patients, 25% patients were in the “low risk” category (329 patients), 49%—in the “moderate risk” category (657 patients), 23%—in the “high risk” category (301 patients), and 3%—in the “critical risk” category (41 patients). The number of observed lethal outcomes was larger in the higher-risk categories, reaching 88% in the “critical risk” category (Figure 1E).

When used for outcome prediction (low or moderate risk—survival; high or critical risk—death), the proposed score showed 55.6% sensitivity and 88.1% specificity.

Each extra five years of delta age adds one point to the score, while each 10 years of chronological age add 2–3 points. The dependency between the score and expected survival time can be expressed as a linear function: Time=242−5×Score ($R^{2}$ = 0.95)—see Appendix A.

In this linear interpretation, each extra five years of delta age subtracts 5 days from the expected survival time of a patient.

## 4. Discussion

In this retrospective study, we demonstrate a model of COVID survival and show that biological aging is a significant factor in COVID-related mortality.

The final model presented here was corrected for seven blood parameters. While considering the ways to define binary features for them, we tried several approaches, including thresholds based on commonly used clinically normal ranges. This approach, however, produced uneven distributions for most blood-related variables. Variables with a high proportion of missing measurements were removed during the first stage of feature selection and only reliable variables were used to create the final model.

Our findings are in agreement with the extensive literature on blood markers in the context of COVID-19. Elevated BUN and creatinine are indicative of renal failure, while LDH increases as a result of organ injury and inflammation [8,9,10,11]. Hyperglycemia and diabetes are also major contributors to poor infection outcomes [12].

One of the markers presented in this study has not been described elsewhere—biological age. COVID’s gerolavic status was evident from the start of the pandemic [13]. The supposed reasons for the elderly being more vulnerable to COVID include being more likely to have multiple comorbidities and weaker immune response [14]. These aspects of aging develop gradually and not necessarily at the same pace in all people. Thus, biological age, as measured by one of the many aging clocks, might be a better determinant of outcome than chronological age alone.

A recent review outlined biological age as a significant contributor to COVID-related mortality, yet did not quantify it with any aging clock [15]. Most aging clocks use hard to obtain molecular-level data (e.g., DNA methylation), but there are also solutions from more routine data types, including clinical blood tests, facial images, surveys [6,16,17].

We chose BloodAge aging clock to measure the pace of aging since it processes the data contained within clinical blood tests collected at patients’ admission (Figure 1A). BloodAge predictions that are higher than chronological age may indicate an increased pace of aging.

Lower delta age in older patients may be interpreted as survivor bias. A higher delta age in dead patients was observed for most age groups and for both sexes (Figure 1B). This indicates that more severe COVID cases either mimic the accelerated aging phenotype or are in part caused by it.

Previously, a study of epigenetic aging clocks concluded that COVID severity is not associated with aging acceleration [18]. The preprint presented on medRxiv compared five COVID-positive patients with ARDS, twelve COVID patients without ARDS, and 17 age-matched controls. COVID patients were predicted younger than the controls on average. The models in this study, however, were not corrected for other possible confounders.

We also observed that crude OR is 1.06 (95% CI: 0.89–1.25) for underagers and is 0.92 (95% CI: 0.79–1.07) for overagers (see Appendix A). These findings are statistically insignificant and thus are not reported in the Results. We consider these results, in aggregation with the non-significance of epigenetic aging for COVID prognostics, an indication of the importance of adjusting for confounders.

We used logistic regression to inspect the effect of accelerated aging separately from other risk factors. In this model, biological age was shown to have double the impact of chronological age on the total mortality rate. In the reported CPH model (Figure 1C), the risk associated with high biological age (HR = 1.026, per year) is of the same magnitude as that associated with chronological age (HR = 1.024, per year).

In another article, the PhenoAge aging clock was used to study the effect of accelerated aging on the infection severity [19]. Akin to BloodAge, PhenoAge uses blood biomarkers to produce a measure of biological age. The authors report AOR for aging acceleration to be 1.50 per five years and for chronological age—1.83 per five years. These figures translate to 1.13 and 1.08 per-year coefficients, respectively. Both BloodAge- and PhenoAge-detected accelerated aging are identified as significant lethal outcome contributors, although their impacts relative to chronological age are different. This may be explained by the differences in the adjustments between the two logistic regressions. Another cause of this behavior is different samples. The PhenoAge study was carried out with a collection of 339,285 people, comprising hospitalized and not hospitalized COVID-positive patients, as well as COVID-negative and untested people. In this large sample, only 613 people were inpatient positives between 16 March 2020 and 17 April 2020; among them, 154 died by 10 January 2020. In comparison, our study was carried out with a sample of 5315 inpatient positives between 3 January 2020 and 12 January 2020; among them, 1689 died by 12 January 2020.

These findings illustrate that biological age may be more informative than chronological age for mortality prediction. The correction for BloodAge may account for individual differences in the aging process and quantify the intuitive understanding of a patient being chronologically old but looking young or the opposite.

In the end, we showed how these findings could potentially be used in a hospital setting by presenting a COVID risk score based on the CPH survival model. The score obtained as a linear combination of mostly binary risk factors can be translated into the expected survival time.

Earlier statistical models predicting patient outcome include an L1-penalized regression, which allows for binary mortality prediction with a sensitivity of 78.0% and a specificity of 87.5% [20]. This model, however, does not allow for time-to-death estimation, was built on a sample half the size of ours, and uses more parameters: 23 compared to 15.

In another work, a logistic regression was used to produce a risk score [21]. That risk score achieved 7.1% sensitivity and 100% specificity in a test sample of 187 patients. The mortality risk score reported here (Figure 1E) yielded 55.6% sensitivity and 88.1% specificity in a larger test set of 1328 patients. Such low sensitivity in our case may be attributed to the assumption that all censored patients were survivors. The actual proportion of lethal outcomes in the sample was probably higher, which masked some true positives as false positives.

Note that since both the risk score and the expected survival time are derived from the survival function, they may be considered as alternative representations of the mortality rate. Unlike another popular COVID risk score, our score requires minimal information about comorbidities and accounts for the pace of aging in patients [22].

## 5. Conclusions

In this study, we have demonstrated the effects of the pace of aging on COVID-related mortality using a CPH survival model. Biological age, as measured by the BloodAge aging clock, was associated with higher mortality risk (HR = 1.026, per year) in the models corrected for chronological age. Lethal cases also showed higher average biological ages than non-lethal cases in all age groups, except for patients older than 80 years.

## 6. Limitations

The models reported focused only on the link between risk factors and all-cause in-hospital mortality. The effect of biological age on infection severity, intubation risk, or need for vasopressor support was not explored.

The grid search for the optimal variables did not exhaust all the possible combinations, and thus a more descriptive survival model with 15 features might exist.

AOR analysis was carried out under the assumption that all censored patients were survivors. The same assumption applied while testing the risk score for outcome prediction.

Finally, there may have been significant differences between the included cohort and the patients that were excluded for having missing parameters needed for calculation of BloodAge; these patients may have been less ill at baseline, leading to, for example, fewer laboratory tests being drawn within 24 h of triage.

## 7. Patents

BloodAge is a patent pending aging clock, see US20200286625A1.

## Figures and Tables

**Figure 1 life-11-00730-f001:**
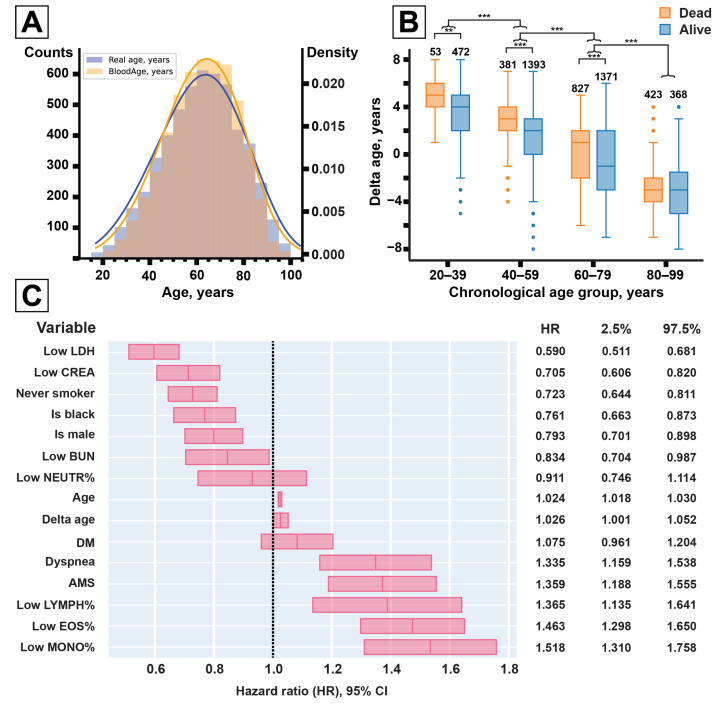
(**A**) BloodAge predictions closely match the real age distribution for the COVID-infected sample (N = 5315 patients). (**B**) Low LDH (HR = 0.59), creatinine (HR = 0.71), BUN (HR = 0.83), and neutrophil count (HR = 0.91) were associated with higher survival time. Low lymphocyte (HR = 1.37), eosinophil (HR = 1.46), and monocyte (HR = 1.52) counts were associated with a shorter survival time. Biological age (Delta age, HR = 1.03, per year) was identified as a significant risk factor, even in the presence of chronological age correction (HR = 1.02, per year). Boxes correspond to the 95% CI of the HRs in the final models. (**C**) BloodAge prediction error (delta age) depends on chronological age in the COVID sample. In all age groups, except for 80–99 years, the lethal cases had higher delta age. The number of patients in each subsample is marked above the box. Boxes represent the interquartile range (IQR) with a median solid line; whiskers extend no further than 1.5 × IQR. Top brackets represent significant U-test results: ** for $<1\times 10^{-4}$, *** for $<1\times 10^{-10}$ (**D**) Survival models can be used as classifiers to predict patient survival in T days (Frame). The classifier derived from the final CPH model reached 62% specificity and 61% sensitivity at T = 18 days (marked with the arrow). (**E**) in the test set comprising 1328 patients, a higher COVID risk score translated into a higher proportion of observed lethal outcomes. Bars are marked with relative proportions in each risk group, the total size of the risk group is marked below the graph (N). AMS = Altered mental state; BUN = blood urea nitrogen; CI = Confidence interval; CPH = Cox proportional hazards model; CREA = creatinine; DM = Diabetes mellitus; EOS% = Eosinophil content; HR = Hazard ratio; LDH = lactate dehydrogenase; LYMPH% = Lymphocyte content; MONO% = Monocyte content; NEUTR% = neutrophil content; P = probability; sensitivity is the proportion of correctly guessed dead patients, specificity is the proportion of correctly guessed living patients.

**Table 1 life-11-00730-t001:** A total of 35,430 unique variable combinations were tested to select the most descriptive variables for the final CPH model. All the variables considered during feature selection are shown in the table below. Variables from different cells of the same group never co-occurred in the same model.

Group	N Options	Alternative Variables				Selected for in Stage:
Race	1	Is black				1
Sex	1	Is male				1
Age	3	Age	Is over 65 years; N years above 65		Is over 65 years	1,2
BloodAge	3	Delta age	Underager; Normal ager; Overager		Aging group	1,2
BMI	3	Is overweight	Is obese		None	1,2
Smoking	3	Never smoker	Ever smoker		Current smoker	1
Symptoms	10	HBP	FeverChills	AMS	Headache	2
		Dyspnea	Cough	GI	Myalgia	
		ChestPain				
		N symptoms				1
History	9	CANCER	CAD	CKD	COPD	2
		CHF	ASTHMA	DM	HTN	
		N comorbidities				1
Blood test	38	Low ALB	Low ALP	Low ALT	Low AST	1,2
		Low BASO%	Low BILID	Low BILIT	Low BUN	
		Low CA	Low CHOLT	Low CL	Low CREA	
		Low EOS%	Low FERR	Low GLC	Low GLOBT	
		Low HCT	Low HDL	Low HGB	Low HGBA1C	
		Low K+	Low LDH	Low LDL	Low LYMPH%	
		Low MCH	Low MCHC	Low MCV	Low MONO%	
		Low MPV	Low NA+	Low NEUTR%	Low P	
		Low PLT	Low PROT	Low RBC	Low RDW	
		Low TRIG		Low WBC		

HBP = High blood pressure; AMS = Altered mental state; GI = Gastro-intestinal disorder; CAD = Coronary artery disease; CKD = Chronic kidney disease; COPD = Chronic obstructive pulmonary disease; CHF = Congestive heart failure; DM = Diabetes mellitus; HTN = Hypertension; ALB = albumin; ALP = Alkaline phosphatase; ALT = Alanine transferase; AST = Aspartate aminotransferase; BASO% = Basophil content; BILD = Direct bilirubin; BILIT = Total bilirubin; BUN = Blood urea nitrogen; CA = Calcium; CHOLT = Total cholesterol; CL = Chloride; CREA = Creatinine; EOS% = Eosinophil content; FERR = Ferritin; GLC = Glucose; GLOBT = Total globulin; HCT = Hematocrit; HDL = High-density lipoprotein; HGB = Hemoglobin; HGBA1C = Glycated hemoglobin; K+ = Potassium; LDH = Lactate dehydrogenase; LDL = Low-density lipoprotein; LYMPH% = Lymphocyte content; MCH = Mean corpuscular hemoglobin; MCHC = Mean corpuscular hemoglobin concentration; MCV = Mean corpuscular volume; MONO% = Monocyte content; MPV = Mean platelet volume; NA+ = Sodium; NEUTR% = Neutrophil content; P = Phosphorus; PLT = Platelet count; PROT = Total protein; RBC = Red blood cell count; RDW = Red blood cell distribution width; TRIG = Triglycerides; WBC = White blood cell count.

**Table 2 life-11-00730-t002:** BloodAge predicts the whole data set and its subdivisions within 6 years of MAE. No significant differences in terms of MAE were detected between the infected and the uninfected cohorts, male and female COVID patients, lethal and non-lethal COVID cases. In terms of mean error, the uninfected patients were predicted to be younger than the infected in non-lethal but not in lethal cases. All metrics were calculated over 100 sampled chronological age-matched cohorts. MAE = Mean Absolute Error; *p*-value (MW) = Mann–Whitney U-test for equal means of the age-matched cohorts; Std = Standard deviation.

Cohort	MAE			Mean Error			N, People
	Years	Std	*p*-Value (MW) ± Std	Years	Std	*p*-Value (MW) ± Std	
Lethal (Total)	2.78	0.01	0.306 ± 0.114	0.43	0.02	*** <0.001	1466
Alive (Total)	2.77	0.02		−0.56	0.04		
Male (Total)	2.74	0.02	0.102 ± 0.069	0.25	0.04	* 0.001 ± 0.002	1723
Female (Total)	2.83	0.01		−0.13	0.02		
Male (Alive)	2.72	0.02	* 0.005 ± 0.007	0.43	0.03	* 0.005 ± 0.008	1159
Female (Alive)	2.83	0.02		0.03	0.02		
Male (Lethal)	2.75	0.03	0.267 ± 0.104	−0.16	0.05	0.267 ± 0.104	513
Female (Lethal)	2.79	0.02		−0.29	0.07		
Lethal (Male)	2.78	0.02	0.052 ± 0.052	0.88	0.03	*** <0.001	922
Alive (Male)	2.64	0.03		−0.10	0.06		
Lethal (Female)	2.77	0.02	0.166 ± 0.117	−0.21	0.04	** <0.001	503
Alive (Female)	2.92	0.05		−1.15	0.08		

*— < 0.01; **— $<1\times 10^{-4}$; ***— $<1\times 10^{-10}$.

**Table 3 life-11-00730-t003:** A total of 15 variables were included into the final survival model, as the result of the grid search.

Variable Name	Variable Description
Age	Continuous chronological age
Is black	The patient stated their ethnicity as “Black” at admission
Is male	The patient stated their sex as “Male” at admission
Never smoker	The patient stated to have never smoked at admission
DM	The patient suffers from diabetes mellitus
AMS	The patient was in an altered mental state at admission
Dyspnea	The patient had shallow breath at admission
Delta age	BloodAge minus chronological age
Low CREA	Creatinine measured at admission ≤84.9 uM
Low BUN	Blood urea nitrogen measured at admission ≤6.43 mM
Low LDH	Lactate dehydrogenase measured at admission ≤441 U/L
Low EOS%	Eosinophil fraction of white blood cells was ≤0.19%
Low LYMPH%	Lymphocyte fraction of white blood cells was ≤13.26%
Low MONO%	Monocyte fraction of white blood cells was ≤6.27%
Low NEUTR%	Neutrophil fraction of white blood cells was ≤77.75%

**Table 4 life-11-00730-t004:** Adjusted odds ratios for the features present in the final survival models. Values in the “Test” column were obtained with a model trained on the whole training set. CV = metric obtained in cross-validation; N = number of patients in a sample; Std = standard deviation across five folds.

	CV	Std	Test
Altered mental state	1.78	0.10	1.78
Age	1.04	0.00	1.04
DM	1.15	0.03	1.15
Delta_age	1.09	0.00	1.08
Dyspnea	1.77	0.07	1.77
Is_black	0.55	0.01	0.55
Is_male	0.76	0.05	0.76
Never_smoker	0.69	0.03	0.69
Low BUN	0.61	0.04	0.61
Low CREA	0.61	0.03	0.60
Low EOS%	1.82	0.06	1.82
Low LDH	0.38	0.01	0.38
Low LYMPH%	2.26	0.22	2.26
Low MONO%	2.25	0.06	2.25
Low NEUTR%	0.80	0.04	0.81
N	3987		1328
Accuracy	0.80	0.01	0.79
Sensitivity	0.61	0.03	0.57
Specificity	0.88	0.01	0.89

**Table 5 life-11-00730-t005:** COVID risk survey that includes BloodAge to estimate a patient’s survival time after summing the points for all the responses. Risk groups are defined in terms of the score: low risk (0–21), moderate risk (22–34), high risk (35–41), critical risk (42–55).

3cPatient’s Chronological Age, Years		
20–29	0	
30–39	2	
40–49	5	
50–59	7	
60–69	10	
70–79	12	
80+	14	
Patient’s BloodAge error		
−10>	0	
−5>	1	
±5	2	
+5<	3	
+10<	4	
Blood parameters		
	Yes	No
BUN ≤ 6.43 mM	0	2
Creatinine ≤ 84.9 uM	0	3
LDH ≤ 441 U/L	0	5
EOS% ≤ 0.19%	3	0
LYMPH% ≤ 13.26%	3	0
MONO% ≤ 6.27%	4	0
NEUTR% ≤ 77.75%	0	1
Other		
	Yes	No
The patient is a never smoker	0	3
The patient has diabetes	1	0
The patient is in an altered mental state	3	0
The patient has dyspnea	3	0
The patient is black	0	3
The patient is male	0	2

## Data Availability

Data were extracted automatically from the Epic electronic medical records (EMR) system. Data used in this study are available on request from authors.

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
