# Peer review of "Increased Pace of Aging in COVID-Related Mortality"

_life, 2021, doi:10.3390/life11080730_

Round 1

Reviewer 1 Report

  • Please, include a background in the abstract, specifying the rationale to carry out the study.
  • Missing information in the abstract: all the methods. The authors have indicated their results at the beginning. Please, read the international standards to report an abstract for an observational study (STROBE).
  • What was your study design? Retrospective cohort study, maybe?
  • “As the same patient may have presented to the ED multiple times, we used only the earliest visit resulting in admission so that each patient contributed unique, non-correlated data” Why not did the authors use models with repeated measures?
  • Why did the authors transform all their continuous predictors into binary variables? Maybe, there are functional forms which would be interesting to be assessed.
  • You should explain the methods beyond klearn.linear_model.
  • Why not using competing risk analysis?
  • How did you develop the score?
  • I would need to know that information before reading the results section. Globally, I think you work is quite interesting, but some particulars are missing so as to understand your manuscript completely.

Author Response

Dear Reviewer 1,

Thank you for taking the time to carefully inspect our manuscript and point out the sections that can be improved.

Please see our point-by-point response to your review.

"As the same patient may have presented to the ED multiple times, we used only the earliest visit resulting in admission so that each patient contributed unique, non-correlated data”
Why not did the authors use models with repeated measures?

As stated in Section 3.1 "Results - Study Sample", only 1.5% of all admitted patients have been admitted more than once.
It is impossible to know the treatment these patients received between discharge and secondary admission. Our database does not allow us to see if these patients recovered and contracted the infection once again or were re-admitted with the prolonged case of their initial affliction.
Thus, we chose the setting in which we could ignore the individual patient effects.

Why did the authors transform all their continuous predictors into binary variables?
Maybe, there are functional forms which would be interesting to be assessed.

The short answer would be: using binarized variables greatly reduces the analysis time and makes the final model more comprehensive. Meanwhile, using continuous variables does not offer significant improvement.
We have tried training the final survival model with all continuous variables. The quality of the model did not significantly improve. In terms of c-index, the continuous version reached an accuracy of 0.752 in training and 0.742 in the test set. Meanwhile, the binarized version reported in the main text displayed the c-index of 0.748 and 0.743, respectively.
We have added these results to Section 3.3 "Biological aging in survival models":
"The model reported reached a c-index of 0.748 in the training set, and 0.743 in the test set. No significant difference in accuracy was detected when all binary variables were switched for their continuous versions: c-index = 0.752 in the training set, c-index = 0.742 in the test set."
To elaborate on why we do not report the models with continuous variables:
First of all, using continuous blood variables rises the problem of scaling. The units of different blood biomarkers are not directly comparable and thus should be scaled to determine which of them is more important. Scaled variables, however, cannot be easily interpreted by humans and require an extra step of preprocessing to assess a patient. This would make both the models and the subsequent risk score harder to use and validate by independent researchers. Thus, we chose median binary transformation as a simple and comprehensive scaling technique. It allows the reader to directly compare the risk associated with high LDH, high creatinine and provides critical cutoff values.
Secondly, we had 38 blood biomarkers to choose from, apart from non-blood variables. To avoid model overfitting we had to perform feature selection and choose only the most relevant features. The resulting combinatorial space was too huge for a complete grid search, so we devised a procedure to sample a fraction of this space and compare the marginal utilities of each variable. The regions we sampled were limited by the rules we derived from similar articles on COVID-19. E.g. all models were adjusted for sex and race. Including these two rules let us reduce the number of models to sample to a quarter of the total number of possible survival models. I.e. we did not have to sample the models that were not adjusted for sex but were adjusted for race etc.
Checking the continuous blood biomarkers alongside their binary forms would have doubled the total number of models to sample. In its turn, this would have greatly increased the computation time. Including the continuous version of non-blood variables would have further increased it.
Please, also note that the main focus of this paper is the pace of aging. To test its impact on the infection outcome, we tried binary, trinary, and continuous variables. Chronological age was also tried as a binary feature, a continuous feature, and a mix of the two. For both chronological and biological aging the continuous forms were the best options. Supplementary materials contain odds ratio values for all forms of age variables.

You should explain the methods beyond sklearn.linear_model

Extended methods are available in the Supplementary Materials, including feature selection, data preprocessing, and classifier construction. We have deliberately moved the full method description to the Supplementary Materials
We have added some brief notes to the main text methods that may help the reader better understand our pipeline.
More specifically,

  • "BloodAge is a deep neural network that was trained to approximate continuous chronological age based on a vector of up to 46 blood biochemical parameters and donor's sex. Its output, compared to the patient's chronological age, represents the intensity of the aging-related changes in a person, compared to their age peers. Higher than chronological age BloodAge values indicate the accelerated pace of aging, while lower -- indicate decelerated aging. (See Supplementary Materials p1)" (Section 2.2);
  • "The survival model was an instance of Cox Proportional Hazards (CPH) implemented with lifelines v0.23.9 for Python. CPH models treat available features as independent risk factors and quantify the probability of an event happening by time t as:..." (Section 2.3)
  • "C-index was defined as the number of pair comparisons in which the model guessed the longer survivor right based on their expected survival time, relative to the total number of all pairwise comparisons." (Section 2.3)
    "Only the variables present in the final survival model were tested." (Section 2.4)
  • "Sensitivity was defined as the number of true positive predictions relative to all positive samples, while specificity -- as the number of true negative predictions relative to all negative predictions." (Section 2.5)

 Why not using competing risk analysis?

Our records do not contain information on the cause of death. Thus, we could not define the competing events and implement this method.

How did you develop the score?

We have included an Excel spreadsheet describing in detail the method briefly outlined in Section 2.6
It lets the reader check how the score will change in response to changing the constants we used.
It also has thirteen mock cases and illustrates how the score translates to the expected time-to-death produced by the CPH model itself.
We have also deployed the alpha version of the COVID risk score at this web address http://44.192.29.144:9028/
You may enter all the values in our model and receive the score itself, the expected TTD, and the survival probability plot.

Please, include a background in the abstract, specifying the rationale to carry out the study.
Missing information in the abstract: all the methods. The authors have indicated their results at the beginning. Please, read the international standards to report an abstract for an observational study (STROBE).

We have reworked the abstract. Having read the STROBE checklist we added the Limitations sections and a new paragraph to the Introduction containing our hypotheses.

We have also corrected some minor grammatical errors in the text.

We hope that these changes to the manuscript answer your questions in full and we can proceed with the review.

Sincerely,
Fedor Galkin

Reviewer 2 Report

Galkin et al. provide an interesting study of COVID-19 and a combination of BloodAge, Age and Risk Factor.

The study is well performed, the results are clear.

It could be accepted after minor revision.

Author Response

Dear Reviewer 2,

We are happy to hear that our manuscript only requires minor revision.

We have corrected the minor language issues you have pointed us towards.

We have also added the Limitations section and included an online COVID risk score calculator you can test here: http://44.192.29.144:9028/

Sincerely yours,
Fedor Galkin

Round 2

Reviewer 1 Report

All my concerns have been correctly addressed, therefore I recommend the publication of this paper in Life.